# A user-centred design framework for mHealth

**Jaydon Farao**[1]*, **Bessie Malila**[1], **Nailah Conrad**[1], **Tinashe Mutsvangwa**[1], **Molebogeng X. Rangaka**[2,3], **Tania S. Douglas**[1]

**1** Medical Imaging Research Unit, Division of Biomedical Engineering, University of Cape Town, Cape Town, South Africa, **2** Division of Epidemiology and Biostatistics, School of Public Health and CIDRI-AFRICA, Institute of Infectious Disease & Molecular Medicine, University of Cape Town, Cape Town, South Africa, **3** Institute for Global Health, University College London, London, United Kingdom

* frxjay002@myuct.ac.za

## Abstract

### Background

Mobile health (mHealth) has the potential to improve access to healthcare, especially in developing countries. The proliferation of mHealth has not been accompanied by a corresponding growth in design guidelines for mHealth applications. This paper proposes a framework for mHealth application design that combines the Information Systems Research (ISR) framework and design thinking. We demonstrate a use case for the proposed framework in the form of an app to read the result of the tuberculin skin test (TST), which is used to screen for latent tuberculosis infection. The framework was used in the re-design of the TST reading app but could also be used in earlier stages of mHealth app design.

### Methods

The ISR framework and design thinking were merged based on how the modes of design thinking integrate with the cycles of the ISR framework. Using the combined framework, we re-designed an mHealth app for TST reading, intended to be used primarily in a developing context by healthcare workers. Using the proposed framework, the app was iterated upon and developed with the aid of personas, observations, prototyping and questionnaires.

### Result

The combined framework was applied through engagement with end-users, namely ten healthcare workers and ten graduate students. Through review of the literature and iterations of the app prototype, we identified various usability requirements and limitations. These included challenges related to image capture and a misunderstanding of instructions. These insights influenced the development and improvement of the app.

### Conclusion

The combined framework allowed for engagement with end-users and for low-cost, rapid development of the app while addressing contextual challenges and needs. The integration of design thinking modes with the ISR cycles was effective in achieving the objectives of each approach. The combined framework acknowledges the importance of engaging users

**Data Availability Statement:** All relevant data are within the manuscript. The summarized and synthesized data included in the paper constitute the minimal data set required to replicate the study's findings.

**Funding:** The research has been supported by the South African Research Chairs Initiative of the Department of Science and Technology and the National Research Foundation of South Africa (Grant No. 98788), and the Center for Innovation in Point-of-Care Technologies for HIV/AIDS at Northwestern University, which is funded by National Institute of Biomedical Imaging and Bioengineering of the National Institutes of Health (Award No. U54EB027049). The content is solely the responsibility of the authors. The funders had no role in study design, data collection and analysis, decision to publish, or preparation of the manuscript.

**Competing interests:** The authors have declared that no competing interests exist.

when implementing mHealth technologies, especially in developing and under-resourced contexts. Findings from this study support the use of this framework as a guide in the design of user-centred mHealth interventions.

## Introduction

A proliferation globally of mobile health (mHealth) applications, which leverage mobile devices for health-related interventions, has been facilitated by high rates of smartphone ownership. In developing countries, smartphone penetration has grown from 21% in 2013 to 53% in 2019 [1, 2]. mHealth apps have the potential to address health-related needs that include disease screening, diagnosis and monitoring. Additionally, mHealth has the potential to alleviate social barriers to healthcare through increased access to health services; reduced health disparities; reduced burden on the healthcare system; and lowered health care costs [3]; this is particularly relevant in low-and middle-income countries.

While the design and the relevance of an mHealth app influence its uptake and ultimately its success, design guidelines are not readily available [4, 5]. If the intention of mHealth interventions is to promote healthy behaviours and facilitate a strengthening of health service delivery, prioritising patient and physician insights and needs becomes integral to the success of the intervention [6], thus engagement with end-users could enable more appropriate app design [7–9]. Furthermore, the context within which mHealth interventions operate, as well as the consequences of implementing the intervention, should be considered in their design [10]. mHealth implementation without end-user engagement, might constrain the desired outcomes, leading to unmet health objectives and possible adverse results [3, 9].

User-centred design is an evidence-based approach that engages with, and prioritises the needs of, end-users during the development of a service or artefact [8, 11]. The World Health Organisation (WHO) endorses this approach and advises that it be integrated within the lifecycle of mHealth interventions, in order to ensure effective outcomes [12], addressing both functionality *and* usability [13]. Thus, both technical and social elements (i.e. how the app fits into the lives of end-users) of the app need to be considered. Currently, few frameworks allow for this holistic approach to mHealth app design in developing and under-resourced contexts. While frameworks are available for the design of mHealth apps [9, 14, 15], few are formulated for a developing context.

The goal of the paper is to explore a combination of user-centred approaches, specifically the Information Systems Research framework and design thinking, for mHealth design in a developing, under-resourced context. We demonstrate the application and evaluation of this framework through the use case of an mHealth app that had been designed to read the results of the tuberculin skin test, which is used to detect latent tuberculosis infection (LTBI), and for which a prototype was available [16]. The framework is thus, in this case, applied to the re-design of an mHealth app. The framework is also intended for use in the early stages of mHealth app design. The proposed methodology centres the user and their health context, emphasising empathy as integral to designing successful mHealth apps.

### Conceptual framework

The proposed approach to designing mHealth apps uses a combination of two existing user-centred design methods, namely the Information Systems Research (ISR) framework and design thinking.

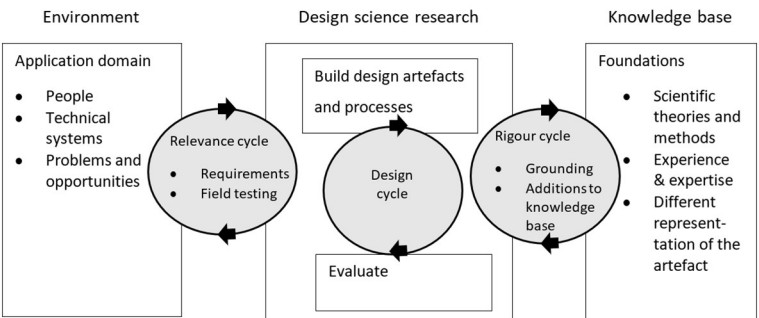

**Fig 1. The Information Systems Research framework.** Republished from [18] under a CC BY license, with permission from Information Systems Research in Scandinavia Association, original copyright 2007.

## The Information Systems Research framework

The ISR framework (Fig 1) has its foundations in design science and behavioural science, and is an information systems tool for innovation [17]. The framework uses iterative, user-centred design processes to create artefacts, usually technological [9]. The ISR framework is comprised of three research cycles: the relevance cycle, where the environment of the end-user is understood; the design cycle, where objects relating to the problem are created and assessed; and the rigour cycle, where findings from the evaluation form part of the existing knowledge base within the problem space [18]. The research cycles link three research domains: the environment, design science research and the knowledge base.

The ISR framework can be applied as an iterative, non-linear process as appropriate to the research context and objectives. Within the framework, a researcher-practitioner relationship is considered. This is how the researchers, who initiate the design project and investigate the knowledge associated with the design, cooperate with the practitioners, who are the professionals working in the design context [19]. These two groups can intersect, but often they are separate and collaborate to design and develop interventions. The practitioner is not necessarily the user but is expected to have knowledge about the context within which an intervention is to be implemented. Schnall et al. [9] considered the ISR framework useful in user-centred mHealth app design, but identified its time-consuming and costly multi-stage approach as an area for improvement. The lack of a definitive design approach to develop artefacts within the design cycle is another limitation.

Various weaknesses of the ISR framework have further been identified, despite its success in practice [19]. In the design cycle, emphasis on the roles of researchers during technical development, is a weakness. There is also a disconnect between the technical requirements and the feasibility of the intervention in the field, and the evaluation of knowledge rigour is not emphasised. The rigour cycle prioritises business objectives over *why* a product works in a given context, which could lead to interventions that work in the short-term but are not sustainable in the long-term.

We propose the use of design thinking within the ISR framework to address these limitations and create a holistic approach for designing mHealth apps.

## Design thinking

Design thinking (DT) applies a user-centred, solution-focused approach to solving complex problems [20]. It is an iterative and interactive process where designing begins with a pictorial depiction of artefacts and evolves into more complex representations as detail is added [21].

Iterations involve the collection of information from end-users, including design specifications and requirements, and using it to inform the modification of the current design [22]. Empathy is a key element of the design thinking philosophy, which is reinforced by improvement of ideas and generating creative concepts through continuous engagement with end-users.

The phases of the design are represented in five modes in DT, as described by the d.school at Stanford University (Fig 2) [23, 24]. The first mode is called *empathise*. This mode represents the initial understanding of the end-user and the problem. It involves a qualitative research approach with observations of end-user behaviour, direct engagement with end-users, and immersing oneself in the experiences of the end-users. This mode provides direction for the subsequent DT modes. The next mode is *define*, which involves the synthesis and analysis of end-user feedback into identifiable needs. Insights are collated and provide perspective for the design challenge. The purpose of the *ideate* mode is to generate as many diverse ideas as possible, which leads to the *prototype* mode where ideas are translated from thoughts to physical representations. Prototyping is meant to be done as quickly as possible with any resources easily available to the designers, ensuring low cost. The final mode is *test*, which involves refining and improving prototypes and simulating their use in the context of the problem. The modes are used non-linearly and iteratively, similarly to the ISR framework, and can be implemented as the designers or researchers see fit. Throughout DT, end-users are engaged to probe the viability of ideas and solutions.

The emphasis on empathy in DT has made it an opportune tool for innovating within clinical contexts, where the needs of patients and healthcare workers have to be prioritised [25]. While the use of DT in healthcare is increasing, many interventions using DT, especially ones that are patient- and health provider-facing, have reported mixed success [25–28]. The use of DT as a design approach in healthcare has challenges, namely: consolidating the needs of end-users, healthcare providers, and researchers; the variations in the approaches taken by

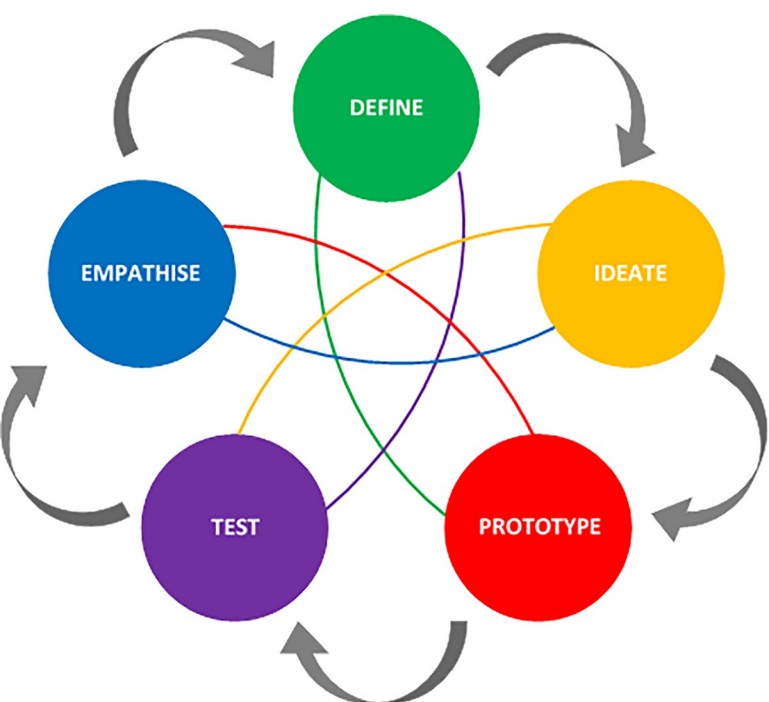

**Fig 2. Design thinking modes, adapted from Reaves [24].**

designers and researchers; and the lack of evidence from studies that include DT as a methodology [25]. The ISR framework can provide some of the necessary tools to alleviate the limitations of DT through its focus on rigorous research and data analysis from various sources, its accommodation of researcher and end-user requirements, and its wide use in other areas of design.

The ways in which DT and ISR can be integrated is apparent when considering the weaknesses both within DT and within the research cycles of the ISR framework, and how each approach has elements that are able to address the weaknesses of the other.

## Combined framework

Table 1 demonstrates how modes of DT address weaknesses within the ISR framework and vice versa, as well as why the modes would integrate for improved user-centred design.

The integration of DT with the ISR framework results in an adapted framework (Fig 3) which we propose for the design of mHealth apps. Based on the nonlinear and iterative nature of DT and the ISR framework [18, 20], both approaches are suited for the design and the re-design of mHealth apps. The combined framework proposed in this study retains these features of the individual approaches. The combined framework is applied here to the re-design of an mHealth app. However, as Michl [29] argues, design may be approached as re-design, in that "design is a sub-category in a never-ending re-design process" and "every complex product that is improved embraces a large number of clever solutions that earlier designers have contributed, and which the later designer freely adopts, makes into his own, and builds on".

**Table 1. Combining design thinking modes with the Information Systems Research framework.**

| Cycle in ISR | Mode/s of design thinking | Why they integrate well |
|---|---|---|
| Relevance | Empathise | • All DT modes and the ISR relevance cycle aim at theorising the requirements for the artefact. Modes have end-user satisfaction and feedback as criteria, while cycles are less defined in this respect. |
| | Define | |
| | | • The collaboration of researcher and practitioner or end-user is emphasised in the modes of design thinking, particularly during the empathise mode. |
| | | • The lack of clarity of acceptance criteria in ISR can be addressed through end-user engagement and empathy, emphasised in DT. |
| Rigour | Test | • The rigour cycle adds to the knowledge base which includes experiences and expertise, as well as existing artefacts and processes. Thus, testing, which includes engagement with end-users (who have relevant experience), and the evaluation of artefacts, suits the function of the rigour cycle. |
| | | • Hevner [18] argues that theory is not enough and that different sources of information e.g. end-users and field testing, assist in providing rigour to the research. |
| | | • Testing can be used by researchers to evaluate the functionalities (and success) of an artefact. Testing can thus be seen as a requirement to meeting health objectives, instead of having the theoretical reasoning for *why* an artefact works competing for significance. By centring the end-user, one can use information gathered during testing as evidence of potential success. |
| Design | Ideate | • The iterative nature of both DT and ISR accommodates the main function of the design cycle. The ideate and prototype modes of DT facilitate the creation of artefacts, a shared characteristic with the design cycle, but quickly and using few resources. The costly and time-consuming nature of ISR framework prototyping is thus alleviated. |
| | Prototype | |
| | | • DT centres the end-user throughout the design process. This allows for participation of practitioners during technical development, a collaboration the ISR does not specify. |

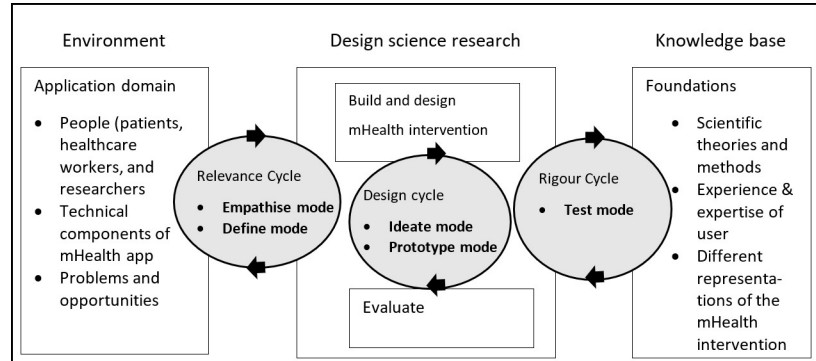

**Fig 3. Modified Information Systems Research framework [18], incorporating modes of design thinking into the relevance, design, and rigour cycles.**

The combined framework can thus be viewed as suitable both for the design and the re-design of mHealth apps.

The combined framework for the design of mHealth apps includes a focus on end-user engagement (while innovation in mHealth has traditionally been technology-driven [4]) and the inclusion of rigour through observations, interviews, and literature to accommodate clinical and end-user needs [30]. Similar work has been done in management studies [31], however, the framework proposed in this paper focuses on mHealth and its application is demonstrated for developing and under-resourced contexts.

Mingers [32] argues that in information systems research, different methods from different paradigms can be combined and used effectively, especially when research situations are complex and multidimensional. In the case of the ISR framework, its roots are in behavioural science and design science. While DT does not have as clear a definition as the ISR framework does, it is derived from design philosophies [31] and has foundations in innovation and user-centred methodologies. Both approaches are concerned with addressing a design problem; however, they do so with contrasting perspectives. The ISR framework as a design science approach has a positivist perspective, in that its knowledge base consists of artefacts and how they function in the real-world based on empirical evidence, and to some extent how generalisable the solutions are [31]. Contrarily, DT has a relativist and constructivist perspective, where the knowledge comes primarily from the design thinker, the end-users, and the context, in order to solve a specific, identified problem [31]. These perspectives, when combined, provide a synergy between the empiricism and rigour of science, and the fluidity and unique nature of design problems.

Embedding DT within the constructs of the ISR framework results in a co-dependent approach that accommodates creativity and creation of artefacts, while developing a knowledge base centred on the end-user, and simultaneously relying on evidence that is primarily obtained from end-users. In mHealth, evidence is important, as interventions can affect the well-being of both patients and healthcare workers while generating new clinical knowledge.

## Ethical considerations

Ethics approval to conduct the research was obtained from the Human Research Ethics Council (HREC) of the Faculty of Health Sciences at UCT (HREC REF: 214/2019), the City of Cape Town Department of Health (ID Number: 8003), and the Western Cape Department of Health (REF: WC_201806_006). Ethics approval was also obtained for the involvement of university

students through the University of Cape Town's Student Affairs office. Informed consent was obtained from all participants.

## Use case: mHealth app to read the TST result

We illustrate the use of the proposed framework through the re-design of an mHealth app to read the result of the Mantoux tuberculin skin test (TST), which is the standard method of detection and early prevention of LTBI [33]. The TST requires patients to visit a clinic for its administration and return to the clinic within 48–72 hours for measurement of the resulting skin induration, if any. This second visit introduces financial and accessibility constraints in developing regions [34]. An mHealth app has been in development to address the constraints related to the TST [16]. The mHealth app proposes an image-based procedure for the analysis of the induration, with image capture carried out by healthcare workers. The previous version of the app required users to capture nine images of the induration, at different angles, after which the images would be processed for analysis. The previous version had various usability limitations, including a complex and difficult to follow interface, which this study sought to address using the proposed framework.

The mHealth app in our use case is intended to be used at a public clinic or during home visits by healthcare workers concerned with TB care. As such, the healthcare workers and their environments became the contexts within which the re-design of the mHealth app took place. The use of the app would occur during the period within which the TST result of a patient is evaluated, and the app would be image-based.

## Relevance cycle

The end-user environment and the information gathered from end-users were captured in the relevance cycle, with the empathise and define modes of DT.

## Empathise mode

To meet the goals of the empathise mode and the relevance cycle, we visited the healthcare facilities within which the mHealth app would be used. These facilities included community health clinics in a township in Cape Town, South Africa. Ten healthcare workers from public clinics, who were involved in the care of patients with tuberculosis, were recruited to participate in the study. They were recruited through purposeful sampling [35]. Ten participants were chosen at each stage of the recruitment process—the mean percentage of problems identified in usability studies rises from 85% to 95% when the number of participants increases from 5 to 10 [36]. Their roles are described in Table 2.

A series of observations and unstructured interviews were conducted with participants in order to understand the end-user's contexts and the specific problems they experienced

**Table 2. Roles and role descriptions of healthcare workers recruited to the study.** Numbers in brackets indicate the number of participants.

| Healthcare worker role (10) | Description |
|---|---|
| Professional nurse (2) | A qualified and competent healthcare worker who can practise comprehensive nursing and midwifery at a professional standard [37]. |
| Assistant nurse (2) | A healthcare worker who provides quality primary nursing care services under the direction of a professional nurse [37]. |
| Community health worker (4) | Members of a community who provide community-based care with a specific focus on HIV and TB care [38]. |
| Counsellors (2) | Health workers that perform HIV and TB counselling, with a focus on adherence and testing assistance [39]. |

relating to the TST. Through these engagements it became apparent that in this under-resourced and oversubscribed healthcare environment, healthcare workers were heavily constrained. The time available for them to engage with the project to re-design the mHealth app using the proposed framework, was limited, and additionally the time they would have available to use such an app in practice was limited.

## Define mode

The feedback from end-users and the observations were collated, synthesised, and thematically analysed [40] to identify the needs of the end-users (clinical and social) as well as the requirements for the mHealth technology. This dual consideration is necessary to deviate from technology-centric solutions without prior end-user engagement. The healthcare worker needs were identified as the requirement for an efficient and comfortable tool with which they could successfully assess and evaluate the results of the TST. Contextual design considerations were identified with which design ideas could be generated for the ideate mode of DT; these included poor eyesight, low literacy, low English language proficiency, an under-resourced environment, and overburdened healthcare workers. The steps for healthcare workers to use the app are summarised in Table 3. The entry point of the app into the TB care cycle would be after an induration forms on a patient's arm (within 48–72 hours of TST administration).

## Design cycle

The goals of the design cycle were met through the ideate, prototype, and test modes of DT. This was especially useful in developing a usable prototype.

## Ideate mode

Ideas were generated in time-constrained intervals, which DT encourages, and represented with sticky-notes and pictorial depictions to accommodate the visual nature of creative DT practices. Some of the ideas formulated included: bilingual interface text, using videos and images for instruction, minimal to no text, and visual directives during image capture. Most ideas were included in the prototypes, as many of the ideas addressed different themes and problems with the interface.

## Prototype mode

The prototype mode of DT usually involves the creation of a low-fidelity prototype, which is iterated upon quickly in order to create a workable, usable, and more defined artefact. This was done in order to understand and explore preferences and the success of the app in achieving functional tasks. Considering the time-constrained and overburdened context of the end-users, we decided to create and iterate prototypes with university students. This allowed for rapid recruitment and swift development of the prototype. The sample group consisted of ten

**Table 3. Requirements for healthcare workers using the mHealth app.**

| Healthcare worker requirements |
| --- |
| 1. Training on image capture and app use |
| 2. Register patient ID on app |
| 3. Explain procedure to patient |
| 4. Capture images of patient's induration on their arm, guided by app |
| 5. Check image quality |
| 6. Save images and communicate next steps with patient |

**Table 4. Participant characteristics during the prototype mode.**

| Participant description | | |
|---|---|---|
| Sex | Male: 5 | Female: 5 |
| Repeat participants | Yes: 5 | No: 5 |
| Low-proficiency English readers | Yes: 0 | No: 10 |
| Comfortable using a smartphone | Yes: 10 | No: 0 |
| **Total** | 10 | |

graduate students, who were recruited using convenience sampling [35]. Their characteristics are described in Table 4. This sample was used to create personas, 'fictitious, specific, and concrete representations' of the intended end-users [41]. Three iterations were carried out with students, in which participants were required to interact with a clickable interface and complete the image capture. Once saturation was reached during each iteration, where no new information was discovered during data analysis and data collection for that iteration, improvements to the interface were made and the next iteration would commence. Observations [42], the "think-aloud" protocol [43], and the post-study system usability questionnaire [44] (a 19-item survey instrument to assess user satisfaction) were used as data collection tools, with the latter used in the final iteration once improvements had been made. Data took the form of assertional analysis of participants' "think-aloud" responses [45]. This type of analysis determines how relationships between concepts were formed during problem-solving through the verbal responses during prototyping, which was appropriate for identifying aspects of the app that needed improvement. Field observation notes were coded manually for patterns associated with the usability of the app and analysed thematically [40, 46].

The clickable prototype represented the mHealth app pages and was designed to address the identified needs, as well as to gain insights for improvement. The third iteration consisted of modifying the Android app based on the feedback obtained. The feedback during this mode came from a non-representative group, thus only served to present ideas for improvement of the app. Each iteration led to improvements which are described in Table 5. The applied changes are

**Table 5. Improvements to identified problems during each iteration.**

| Iteration | Problems identified | Changes implemented |
|---|---|---|
| First | • Difficulty following and remembering instructions | • Increasing the size of the text |
| | | • Including single page instructions |
| | • Lengthy text | • Pictorial instructions |
| | • Complex language for low-proficiency English readers and low literacy users. | • Removal of jargon and technical terms used in the app |
| | | • Adding videos to describe the image capture process |
| Second | • Unclear video instructions without audio assistance | • Colour scheme changes |
| | | • Font changes |
| | • Difficulty capturing all images at good quality | • Adding audio to the video instructions |
| | • Difficulty differentiating colours of text and background | • Scrolling icon to indicate a scrollable page |
| | | • Decreasing the text on the instruction page |
| | • Misunderstanding of continuous, scrollable app pages and disjointed ones | |
| Third (Where findings from the previous iterations were implemented in the mobile app as a high-fidelity prototype) | • Difficulty capturing multiple images at different angles | • Adding additional guidance when capturing images at different angles |
| | • Uncertainty when submitting images | • Allowing submission of images with ease through clear presentation buttons and instructions |

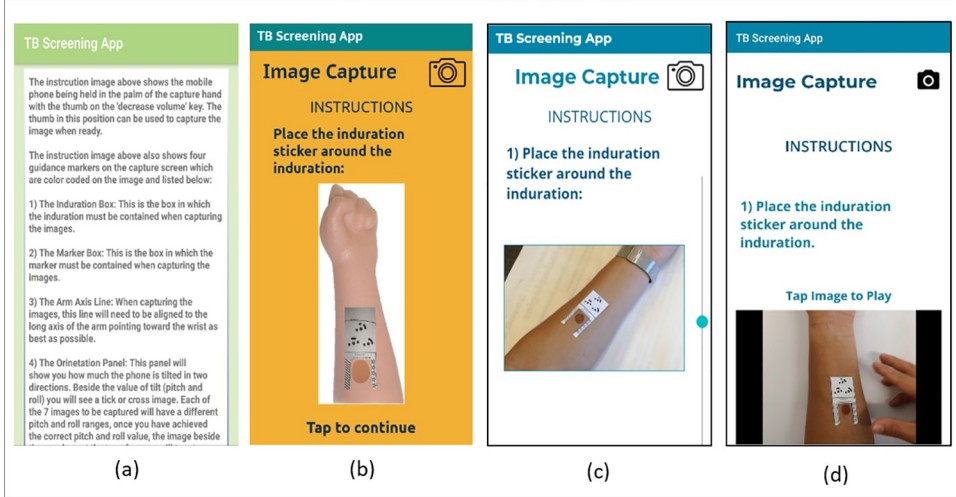

**Fig 4.** Prototype progression of the instruction pages of the mobile app; (a) is the previous interface, with progression through (b) and (c) leading to (d), the updated interface.

represented as screenshots (Fig 4) of the instruction page progressing from the original interface (a) to the updated interface (d). The final prototype is shown in Fig 5. During prototyping, student participants were required to capture images of simulated indurations on their own arms, as this was considered more challenging than capturing it on someone else's arm (the case with healthcare worker participants). Designing in this way, where user needs are amplified, allow for better understanding of the needs of users that fall outside the ideal case [23].

The changes to the final prototype allowed for an app version that was ready for testing with intended end-users, the healthcare workers at an under-resourced and overburdened clinic involved in TB care. Additionally, the app was translated into isiXhosa, the dominant language of the healthcare workers. This stage of the design initiated the rigour cycle of ISR and the test mode of DT.

## Rigour cycle

The rigour cycle, where findings from the evaluation form part of the existing knowledge base, was divided into two phases. The first phase was the reviewing of the existing literature related

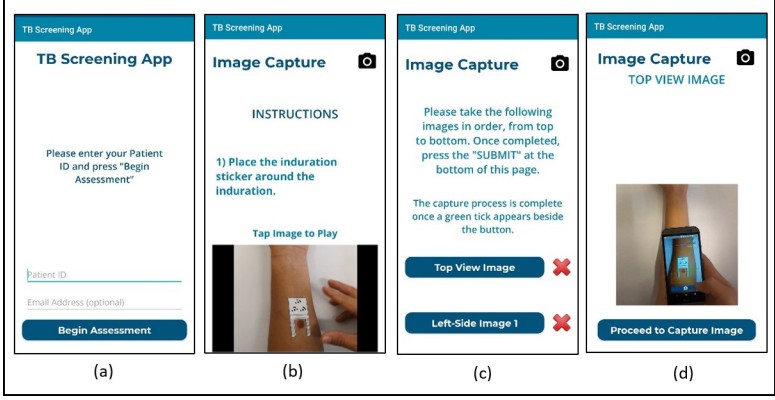

**Fig 5.** Final iteration prototype screenshots—a) landing page, b) instruction page, c) image capture page, d) image capture instruction video.

to the problem and the context. This provided the preliminary information for the knowledge base and a foundation with which to develop ideas. The second phase was the testing of the prototype, which constitutes an evaluation of the idea in the appropriate context. This adds to the knowledge base and informs further steps towards the design of the mHealth app. These two phases of the rigour cycle occur at different stages of the design due to the iterative nature of the proposed framework. The first phase of the rigour cycle therefore began with a review of the literature to identify usability considerations when designing mHealth apps. This cycle took place before the relevance cycle, and the findings influenced some of the design decisions made during app development. The review indicated that the context within which an app is used affects usability and quality of mobile apps [47], usability is primarily influenced by the user-interface [48], and clarity of instructions, which affects time to learn, is necessary for designing mobile apps [49]. The ISR framework was applied iteratively, and once the prototype mode within the design cycle was complete, the test mode was implemented.

## Test mode

The healthcare workers who were considered during the empathise mode (Table 2) formed the sample group (N = 10) during the test mode. Their characteristics are described in Table 6. The same data collection tools used during prototyping (think aloud, observations, and PSSUQ) were applied. Assertional analysis [45] was also applied to "think-aloud" responses. Additionally, participants were asked whether they were comfortable using a smartphone (all were comfortable), and which language they would prefer the app to be in (four community health workers preferred isiXhosa, while the rest of the participants preferred English).

Some of the common problems that arose during the test mode included:

- not spending time watching the instruction videos;

- incorrect navigation while progressing through the app pages;

- incorrect distance of the phone from the induration site when capturing images; and

- rushing through the final images without much regard for image quality.

Healthcare worker participants expressed frustration with the long duration of image capture. However, many participants commented that they "just need time" to learn. Considering

**Table 6. Participant characteristics during the test mode.**

| Participant category | Responses | | |
|---|---|---|---|
| Counsellors | Sex | Male: 0 | Female: 2 |
| | App language | isiXhosa: 0 | English: 2 |
| | Comfortable using a smartphone | Yes: 2 | No: 0 |
| Professional Nurses | Sex | Male: 1 | Female: 1 |
| | App language | isiXhosa: 0 | English: 2 |
| | Comfortable using a smartphone | Yes: 2 | No: 0 |
| Assistant Nurses | Sex | Male: 0 | Female: 2 |
| | App language | isiXhosa: 0 | English: 2 |
| | Comfortable using a smartphone | Yes: 2 | No: 0 |
| Community Health Workers | Sex | Male: 0 | Female: 4 |
| | App language | isiXhosa: 4 | English: 0 |
| | Comfortable using a smartphone | Yes: 4 | No: 0 |
| Total | | 10 | |

the needs of healthcare workers, identified during the relevance cycle and define mode, the participants were satisfied with the app and its accommodation of their basic needs with respect to the TST. However, needs arose that are unrelated to the TST, but are related to the impact of the mobile phone-based app on their daily lives.

Additional comments came from the community health workers (CHWs) who raised questions on the practicality of the app in their contexts. This was related to their mobility when carrying out their clinical work, and the possibility of a mobile phone-based intervention introducing threats to their safety as the mobile device would be a target for theft. Questions of data safety also arose, relating to where the captured data would be saved and who would see it. These questions were addressed by describing who would interact with the personal medical data in the public health context (nurses, CHW, doctors, and researchers).

Healthcare worker participants captured images of a simulated induration on a research assistant's arm, representing the ideal use case for the app.

## The post-study system usability questionnaire (PSSUQ)

Ten student participants and ten healthcare worker participants completed the PSSUQ which required them to rate different aspects of usability on a scale from 1 (strongly agree) to 7 (strongly disagree). The questions relate to three factors affecting usability: system usefulness–the ease of use and learning, the speed at which users become productive, and the effective completion of tasks; information quality–the feedback the system provides related to error messages and fixing problems, how well the information is organised, whether it is easy to understand, and whether it helps the user complete tasks; and interface quality–the affability of the system for the user, as well as the likeability of the system [50]. The average scores for both participant groups were calculated and presented as a "level of agreement" (Fig 6).

Both participant groups performed similarly with an average score below three for all usability factors. Thus, the satisfaction with the aspects of the app tested in the questionnaire was neutral to strong [50]. Additionally, comments were left on the questionnaire relating to participant experiences using the app. Many comments concerned the learnability of the app and how repeated attempts would result in better and quicker completion of the image capture. Both participant groups also noted that training would be necessary for accurate and successful progress through the app. The interface was described as visually pleasing, while the

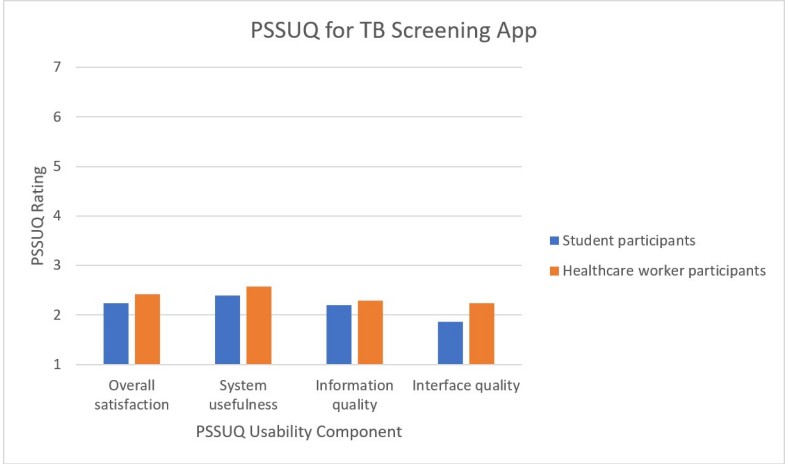

**Fig 6. Responses to the PSSUQ by student and healthcare worker participants.**

**Table 7. Overview of methods, analysis, and sampling techniques used in the application of the combined framework.**

| Cycle in ISR | Mode/s of design thinking | Methods and analysis | Sampling |
|---|---|---|---|
| Relevance | Empathise | Observations and unstructured interviews | Purposeful |
| | Define | Thematic analysis | |
| Rigour | Test | Observations; "think-aloud" protocol; post-study system usability questionnaire. | Purposeful |
| | | Assertional and thematic analysis. | |
| Design | Ideate | Brainstorming | |
| | Prototype | Low-fidelity, iterative prototyping; personas; observations; "think-aloud" protocol; post-study system usability questionnaire. | Convenience |
| | | Assertional and thematic analysis. | |

major usability concerns related to the long and challenging image capture. Video and audio guidance were recognised as helpful, but the number of images required remained a challenge.

Finally, to summarise, Table 7 presents the various methods and sampling used during each cycle and mode of the ISR and design thinking.

## Discussion

The integration of the ISR framework and design thinking (Fig 3) has been illustrated for the re-design and improvement of an mHealth app aimed to read the result of the TST for latent tuberculosis infection. This combined framework introduces a means both to achieve usable mHealth apps and to accommodate the environments and challenges that end-users face in developing and under-resourced contexts.

The proposed framework combined the iterative and creative nature of design thinking with the rigour and structure of the ISR framework. The modes of design thinking facilitate the generation of ideas and the creation of low-fidelity prototypes. Through the visualisation of the interface of the app with a low-fidelity prototype, student participants could interact with the representation and identify usability concerns early in the design. They could also do this rapidly and at little to no cost to the designers. Healthcare workers could then identify how context-specific limitations would influence app usability. Time spent with healthcare workers was constrained by their busy schedules, but through the design process suggested in this paper, most of the usability limitations could be mitigated in the prior engagement with students. Despite the prior engagement and improvements with students, however, usability constraints arose with healthcare workers. Iterating with students before involving healthcare workers allowed for quick and efficient refinement of the user-interface, and further allowed for more meaningful interactions with end-users in the field. Given the above considerations, we believe that the design approach described in this paper addresses the cost and time concerns associated with the application of the ISR framework highlighted by Schnall et al. [9]. Additionally, using design thinking, especially iterative prototyping, as a technique for artefact development during the design cycle proved beneficial for creating a diversity of prototypes and stimulating creativity.

Each mode of design thinking integrated well with the cycles of the ISR framework (Table 1) to achieve the objectives of each cycle and mode. The application of a DT approach encouraged creativity in the development of the mHealth app and enabled an empathetic and reflexive attitude, which was important since the researchers involved did not necessarily share characteristics with end-users. This empathetic and reflexive approach supports mHealth interventions that can integrate with the existing healthcare system which ultimately allows for easier adoption of these technologies.

## Relevance cycle including the empathise and define mode

The inclusion of the empathise and define modes within the relevance cycle, and a consideration of empathy throughout the re-design process, was enabled through a contextual understanding of users, their needs, and their environment. It allowed for identification and assessment of needs. Further development of the framework could include the exploration of additional design thinking techniques [23], such as a 'user camera study', which uses photographs to better understand users' lives and specific tasks in their contexts, and a 'composite character profile', for observing relevant characteristics of an end-user. Both techniques could assist in designing mHealth apps with empathy in unfamiliar contexts.

## Rigour cycle including the test mode

The test mode allowed interactions with end-users that provided evidence of the potential success of the mHealth app in the context of a primary healthcare facility. Combining the insights learned through testing with those obtained from the literature, enhances the objective of the rigour cycle, as it considers how the contextual circumstances might affect the design of the mHealth tool. This consideration of context has the potential to create more sustainable interventions in under-resourced settings. One of the dominant concerns was the time required to implement the app by end-users in healthcare. Further work could include innovations in testing, such as remote testing (away from the clinic) and feedback on the prototype in the end-user's own time using digital technologies.

## Design cycle including the ideate and prototype mode

The ideate and prototype modes encouraged creative thinking, which is suggested by the objectives of the ISR design cycle. A limitation of the framework as applied in this study was a low number of participants in the ideate mode, as diversity and quantity of ideas is beneficial to idea development. This limitation should be balanced with the cost and time involved in engaging large groups to provide ideas. Furthermore, since ideation and prototyping can take many forms, creativity in this respect is encouraged and future work could explore the use other DT techniques [23], including 'impose constraints', which involves intentionally adding restrictions to how one ideates and prototypes. In doing so, creative potential can increase. In an under-resourced clinical setting, this could be effective since the context may have constraints the researchers are unaware of.

In the clinical environment, the usability considerations that arose became more contextual and specific to healthcare worker roles. For example, community healthcare workers were concerned about their personal safety. This would affect the usability of the app since healthcare workers may be deterred from using the app if they are vulnerable to theft and other threats. Coetzee et al. [51] report concerns of safety in a Cape Town township for community healthcare workers who carry devices. Hussain and Kutar [52] include safety as a component within their usability guidelines for mobile apps. Through both the empathy mode of DT and consulting relevant literature, these usability concerns could be identified. The usability considerations and needs identified during initial engagement with healthcare workers were addressed by the basic functionalities of the app; however, through further engagement, additional considerations and needs arose, highlighting the importance of continuous reflexive engagement with end-users.

In addition to safety concerns, Labrique et al. [53] emphasise that confidentiality of data is compromised should a phone with sensitive information be stolen. Data confidentiality should be a usability consideration when designing an app, since usability and trust is vital when attempting to implement electronic health interventions [54]. Communicating the flow of

personal medical data, and security thereof, is important to both healthcare workers and the patients with whom they engage. The concerns raised by healthcare workers related to health information and access to it, can affect the adoption of an mHealth app. Engaging with users empathetically and exploring DT techniques that allow holistic and extensive communication could benefit the reception of an intervention by the end-user group.

Even though there are benefits to engaging with two different user-groups, the design was limited because end-users (healthcare workers) were not engaged throughout; this was the result of the constraints on their time and their availability to participate in a design process while working in an over-burdened health system. Additionally, the small number of participants limits generalisability of the results. However, a sample size of 10±2 is sufficient for discovering 80 per cent of usability issues [55]. Testing in diverse contexts would be required if the mHealth app is to be scaled across more diverse health settings and user groups.

The use of the PSSUQ in English was a limitation with a subgroup of users who preferred isiXhosa as their spoken language and language of instruction. Efforts were made to simplify some of the terminology of the questionnaire; however, there remains the possibility of misunderstanding some of the questions. A validated usability evaluation tool that is context-specific and translated could be further investigated as a way of further centring the user throughout the design and evaluation of mHealth apps. This would enrich the combined framework and extend its user-centred approach.

The mHealth app used in this study is intended for use as a tool in screening for latent tuberculosis infection. The app considers healthcare workers as end-users. The proposed framework would be appropriate for similar mHealth interventions. However, the expansion of the app for a diversity of applications, including consumer-based and self-monitoring mHealth apps, would require some context-specific adaptation. The emphasis of the framework on user-centred design is an advantage and is broadly applicable to mHealth, given recent recognition of the benefits of end-user engagement and feedback during the mHealth lifecycle [56, 57]. The proposed framework has been applied to the re-design of an mHealth app, in what can be considered an iteration of an existing app. While the study focuses on this single use case, the framework is also suitable for the initial design of mHealth apps since both DT and the ISR framework are iterative and non-linear, and allow for the designer's discretion when initiating the design process, based on the needs of the project and end-user [18, 20]. The framework is especially well-suited for mHealth apps that are situated within constrained and under-resourced contexts.

Interventions using design thinking in healthcare have usually been tested in industrialized countries, even more so for mHealth apps. The proposed framework presents an opportunity to improve context-specific interventions, focusing on empathy and reflexivity, as well as to provide more evidence of design methods that are suitable for developing, marginalised, and under-resourced contexts.

## Conclusion

We have presented a framework for the design of mHealth apps that combines the Information Systems Research Framework and design thinking. The use of the combined framework was demonstrated in the re-design of an mHealth app to read the result of the TST for LTBI screening. The combined framework centred the user holistically throughout the design of the app, and its iterative nature allowed for rapid improvements that incorporated user feedback. The framework may be used to guide future user-centred development of mHealth apps, especially in developing and under-resourced contexts.

## Author Contributions

**Conceptualization:** Tinashe Mutsvangwa, Molebogeng X. Rangaka, Tania S. Douglas.

**Data curation:** Jaydon Farao.

**Formal analysis:** Jaydon Farao.

**Investigation:** Jaydon Farao, Bessie Malila.

**Methodology:** Jaydon Farao, Nailah Conrad.

**Project administration:** Bessie Malila, Tania S. Douglas.

**Software:** Jaydon Farao.

**Writing – original draft:** Jaydon Farao.

**Writing – review & editing:** Jaydon Farao, Bessie Malila, Nailah Conrad, Tinashe Mutsvangwa, Molebogeng X. Rangaka, Tania S. Douglas.

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
