## [Decision Letter · Decision Letter 0]

25 Mar 2020

PONE-D-20-01209

A user-centred design framework for mHealth

PLOS ONE

Dear Mr. Farao,

Thank you for submitting your manuscript to PLOS ONE. After careful consideration, we feel that it has merit but does not fully meet PLOS ONE’s publication criteria as it currently stands. Therefore, we invite you to submit a revised version of the manuscript that addresses the points raised during the review process.

I have carefully reviewed the manuscript as well as the reviewers' comments and in addition to their concerns, I have a number of my own concerns which would need to be addressed to ensure that this manuscript is suitable for publication. First, the goal of the paper is unclear -- is this to present a new framework for the design of mHealth applications and if so then the rigor around the development of this framework is needed. Further, the issue described as cited by schnall et al. with the ISR framework is an time consuming and cost intensive process. Yet, its unclear how the revised framework addresses these issues. In addition, I think its important that the authors carefully consider the reviewers' thoughtful and constructive comments.

We would appreciate receiving your revised manuscript by May 09 2020 11:59PM. To enhance the reproducibility of your results, we recommend that if applicable you deposit your laboratory protocols in protocols.io, where a protocol can be assigned its own identifier (DOI) such that it can be cited independently in the future. For instructions see: http://journals.plos.org/plosone/s/submission-guidelines#loc-laboratory-protocols

We look forward to receiving your revised manuscript.

Kind regards,

Rebecca Schnall

Academic Editor

PLOS ONE

Journal Requirements:

3. We note that Figure(s) [#] in your submission contain copyrighted images. All PLOS content is published under the Creative Commons Attribution License (CC BY 4.0), which means that the manuscript, images, and Supporting Information files will be freely available online, and any third party is permitted to access, download, copy, distribute, and use these materials in any way, even commercially, with proper attribution. For more information, see our copyright guidelines: http://journals.plos.org/plosone/s/licenses-and-copyright.

1.    You may seek permission from the original copyright holder of Figure(s) [#] to publish the content specifically under the CC BY 4.0 license. 

Reviewers' comments:

Reviewer's Responses to Questions

**Comments to the Author**

1. Is the manuscript technically sound, and do the data support the conclusions?

Reviewer #1: Partly

Reviewer #2: Partly

2. Has the statistical analysis been performed appropriately and rigorously? 

Reviewer #1: No

Reviewer #2: No

3. Have the authors made all data underlying the findings in their manuscript fully available?

Reviewer #1: No

Reviewer #2: Yes

4. Is the manuscript presented in an intelligible fashion and written in standard English?

Reviewer #1: Yes

Reviewer #2: Yes

5. Review Comments to the Author

Reviewer #1: The idea of the paper is very interesting as the use of a mobile app in healthcare has rapidly increased and the evidence-based framework should be used to guide its design. However, there are several concerns in this study.

1. It is not clear if the approach authors have focused on is “user-centered design” or “participatory design” although they are different. It should be clearly differentiated in the introduction section and followed by that, the introduction should be re-written as it is not consistent in this manuscript.

2. The first and second paragraph should be connected. The potential of mHealth interventions in developing countries should be clear; the authors described that mHealth has the potential to increase access to health services, then mentioned it hasn’t been accompanied by the positive clinical outcomes? In the next paragraph, again to promote healthy behaviors, and then prioritizing patient and physician insights and needs? What is the success of mHealth interventions? More appropriate than what? The importance of the study should be well-stated in the introduction.

3. The authors have claimed that “Currently, few frameworks allow for this holistic approach to mHealth app design.” To my knowledge, there are great approaches which can be used for the design of mHealth apps. Existing literature of the mHealth app design approach, usability of the mHealth interventions and evidence-based frameworks which have been used in this area should be elucidated, a problem to be addressed should be stated, and then an important need of the authors’ new framework should be proposed with the clearly stated purpose of the study.

4. The conceptual framework could be well-summarized in the introduction rather than the methods section, with the further rational for the combined framework, if the authors aimed to evaluate the applicability of the combined framework to their mHealth app. Or, if the purpose of the study was to report the approach, it would be good to make the manuscript containing a case study. Once again, the purpose of the study in this manuscript should be clearly stated.

5. The rational of the sample size and data analysis should be added.

6. Each cycle within the proposed framework could include methods (study sample/procedures/data analysis) and results independently, which will enable the readers to avoid the confusion and enhance an understanding of using the framework for the mHealth app design. Moreover, it would be good to create a table displaying the methodologies used for each cycle/mode (e.g., think-aloud protocols/ PSSUQ).

7. Tables of the study sample characteristics and the results of the PSSUQ should be added.

8. In the discussion section, the key findings for EACH cycle/mode could be discussed as this was to propose an approach. Moreover, the most appropriate approach in mHealth apps’ design would differ in a type/aim of mHealth apps (e.g., consumer app/ screening app), and actual end-users using the apps (e.g., patients/ health professionals), and the authors could add the in-depth discussion to set a boundary for using the combined framework for mHealth.

9. Overall, this manuscript appears to be verbosely written, and the purpose of the study as well as type of study are not clearly stated.

Reviewer #2: The manuscript titled “A User-centered design framework for mHealth” submitted by Jaydon Ethan Farao, MPhil et al. presents a framework of design for mHealth studies. The framework combines the information systems research and design thinking and contains three cycles: relevance, design, and rigour cycle. They especially emphasize the empathise mode and state that both functionality and usability should both be addressed in the relevance cycle. A case study is discussed in the paper to illustrate how the proposed theory was used to guide the design of an mHealth app for reading TST results. Overall, I think the authors are working on a very important and interesting research problem, which could have potential impacts on mHealth studies; and they explain the theory of the proposed framework pretty well. My major comments are:

1. I feel like more quantitative evidence is needed to make it convincing that the proposed framework could be widely applicable, for example, data to show its advantages over traditionally used design methods.

2. For the case study, would the author be able to provide more detailed data on the participants’ needs that you identified in the first cycle and how they felt if their needs have been met when they tested the designed your app.

3. Could you explain why did not your include literature review in the relevance design?

4. Label of y-axes of figure 6 needs to be added.

6. PLOS authors have the option to publish the peer review history of their article (what does this mean?). If published, this will include your full peer review and any attached files.

Reviewer #1: No

Reviewer #2: No

---

## [Author Response · Author response to Decision Letter 0]

11 May 2020

We thank the editor and the reviewers for their valuable comments. We explain in the attachment titled “Response to Reviewers” how we have addressed them to improve the clarity of the manuscript.

---

## [Decision Letter · Decision Letter 1]

24 Jun 2020

PONE-D-20-01209R1

A user-centred design framework for mHealth

PLOS ONE

Dear Dr. Farao,

Thank you for submitting your manuscript to PLOS ONE. After careful consideration, we feel that it has merit but does not fully meet PLOS ONE’s publication criteria as it currently stands. Therefore, we invite you to submit a revised version of the manuscript that addresses the points raised during the review process.

There are some very minor revisions that are still needed for this manuscript. Please see the comments from reviewer #1.

We look forward to receiving your revised manuscript.

Kind regards,

Rebecca Schnall

Academic Editor

PLOS ONE

Additional Editor Comments (if provided):

The reviewers considered your revised manuscript to be very responsive to the earlier reviews. There are some minor revisions which are still recommended by Reviewer 1.

Please address.

Reviewers' comments:

Reviewer's Responses to Questions

**Comments to the Author**

1. If the authors have adequately addressed your comments raised in a previous round of review and you feel that this manuscript is now acceptable for publication, you may indicate that here to bypass the “Comments to the Author” section, enter your conflict of interest statement in the “Confidential to Editor” section, and submit your "Accept" recommendation.

Reviewer #1: All comments have been addressed

Reviewer #2: All comments have been addressed

2. Is the manuscript technically sound, and do the data support the conclusions?

Reviewer #1: Yes

Reviewer #2: Yes

3. Has the statistical analysis been performed appropriately and rigorously? 

Reviewer #1: Yes

Reviewer #2: Yes

4. Have the authors made all data underlying the findings in their manuscript fully available?

Reviewer #1: Yes

Reviewer #2: Yes

5. Is the manuscript presented in an intelligible fashion and written in standard English?

Reviewer #1: Yes

Reviewer #2: Yes

6. Review Comments to the Author

Reviewer #1: The revised manuscript has been improved by addressing reviewers’ comments. There are two additional comments:

1) In the Abstract, the Results could be improved. Some appear to be discussion/conclusion rather than results.

2) While it says the purpose of the study is to propose a framework for the design of an mHealth app, it appears that the authors actually used the combined framework to RE-DESIGN an mHealth app for TST reading, and it was ONE case study. As the most appropriate methodological approach would differ in mHealth apps’ RE-DESIGN than in mHealth apps’ initial design process, it would be good to clarify within the entire manuscript.

Reviewer #2: (No Response)

7. PLOS authors have the option to publish the peer review history of their article (what does this mean?). If published, this will include your full peer review and any attached files.

Reviewer #1: No

Reviewer #2: No

---

## [Author Response · Author response to Decision Letter 1]

17 Jul 2020

The response to reviewers is detailed in the accompanying document titled "Response to Reviewers".

---

## [Editor Report · Decision Letter 2]

6 Aug 2020

A user-centred design framework for mHealth

PONE-D-20-01209R2

Dear Dr. Farao,

We’re pleased to inform you that your manuscript has been judged scientifically suitable for publication and will be formally accepted for publication once it meets all outstanding technical requirements.

Kind regards,

Rebecca Schnall

Academic Editor

PLOS ONE

---

## [Editor Report · Acceptance letter]

10 Aug 2020

PONE-D-20-01209R2 

A user-centred design framework for mHealth 

Dear Dr. Farao:

I'm pleased to inform you that your manuscript has been deemed suitable for publication in PLOS ONE. Congratulations! Your manuscript is now with our production department. 

Kind regards, 

on behalf of

Dr. Rebecca Schnall 

Academic Editor

PLOS ONE